# Strength Enhancement of Regenerated Cellulose Fibers by Adjustment of Hydrogen Bond Distribution in Ionic Liquid

**DOI:** 10.3390/polym14102030

**Published:** 2022-05-16

**Authors:** Yu Xue, Weidong Li, Guihua Yang, Zhaoyun Lin, Letian Qi, Peihua Zhu, Jinghua Yu, Jiachuan Chen

**Affiliations:** 1School of Chemistry and Chemical Engineering, University of Jinan, Jinan 250022, China; xxyy0707@163.com (Y.X.); chm_zhuph@ujn.edu.cn (P.Z.); chm_yujh@ujn.edu.cn (J.Y.); 2State Key Laboratory of Biobased Material and Green Papermaking, Qilu University of Technology, Shandong Academy of Sciences, Jinan 250353, China; 17854116982@163.com (W.L.); linzhaoyun@qlu.edu.cn (Z.L.); chenjc@qlu.edu.cn (J.C.)

**Keywords:** regenerated cellulose fiber, hydrogen bond, physical strength, ionic liquid, additive

## Abstract

To improve the physical strength of regenerated cellulose fibers, cellulose dissolution was analyzed with a conductor-like screening model for real solvents in which 1-allyl-3-methylimidazolium chloride (AMIMCl) worked only as a hydrogen bond acceptor while dissolving the cellulose. This process could be promoted by the addition of urea, glycerol, and choline chloride. The dissolution and regeneration of cellulose was achieved through dry-jet and wet-spinning. The results demonstrated that the addition of hydrogen bond donors and acceptors either on their own or in combination can enhance the tensile strength, but their effects on the crystallinity of the regenerated fibers were quite limited. Compared with the regenerated fibers without any additives, the tensile strength was improved from 54.43 MPa to 139.62 MPa after introducing the choline chloride and glycerol, while related the crystallinity was only changed from 60.06% to 62.97%. By contrast, a more compact structure and fewer pores on the fiber surface were identified in samples with additives along with well-preserved cellulose frameworks. Besides, it should be noted that an optimization in the overall thermal stability was obtained in samples with additives. The significant effect of regenerated cellulose with the addition of glycerol was attributed to the reduction of cellulose damage by slowing down the dissolution and cross-linking in the cellulose viscose. The enhancement of the physical strength of regenerated cellulose fiber can be realized by the appropriate adjustment of the hydrogen bond distribution in the ionic liquid system with additives.

## 1. Introduction

Plants can produce more than 1.5 × 10^12^ tons of cellulose every year [1], which could be widely applied in the fields of paper-making, textiles, food, and medicine industries. The presence of hydrogen bonds in cellulose structures makes it difficult to dissolve within water and other ordinary solvents, unless harsh treatment was employed [1]. The existing traditional approaches for cellulose processing are usually performed under some severe treatment conditions such as strong acid and alkaline that may cause numerous drawbacks, including chemical pollution, excessive energy consumption, and complicated procedures.

In general, ionic liquids (ILs) can dissolve cellulose by disturbing the native hydrogen bond network within cellulose structures [2]. In previous studies, imidazolium-based ILs, such as 1-butyl-3-methylimidazolium chloride (BMIMCl) [3,4], 1-allyl-3-methylimidazolium chloride (AMIMCl) [5], 1-butyl-3-methylimidazolium acetate [6,7], and 1-ethyl-3-methylimidazolium acetate [8], have been broadly utilized as favorable cellulose solvents. The cellulose solubility of IL is strongly dependent on the properties of anions and cations. It was found that both the anions and cations of the ILs participate in the cellulose dissolution, and thereby the imidazolium received tremendous attention due to various advanced developments [9]. The anions play dominant roles in the cellulose dissolution within ILs. The strong electronegativity in IL anions allows them to form strong hydrogen bonds with the hydroxide groups in cellulose, thus leading to the breakage of the native cellulose hydrogen bond network. Chloride anion-based ILs are widely adopted for cellulose pretreatment, as the anion is a small-sized hydrophilic hydrogen bond acceptor, which facilitates the occurrence of cellulose dissolution [9]. AMIMCl belongs to one of the most effective ILs for cellulose dissolution and regeneration because of its high chloride anion concentration [10], efficient synthesis, nonvolatility, nonflammability as well as chemical and electrochemical stability [10].

The dissolved cellulose can be spun to fabricate the regenerated cellulose fibers with excellent tensile strength (1.6–5.34 cN/dtex), which can be comparable to Lyocell fibers, to meet the demands of daily living [11]. Additives are commonly used to improve the quality of regenerated cellulose fibers. Meanwhile, co-solvents applied in imidazolium-based ILs can further promote the cellulose dissolution and enhance the spinnability of cellulose viscose. Minnick [12] added dimethyl sulfoxide, dimethylformamide, and 1,3-dimethyl-2-imidazolidinone into imidazolium-based ILs to upgrade their solubility. Additionally, it demonstrated that the introduction of 1,3-dimethyl-2-imidazolidinone into BMIMCl can reduce the viscosities [13]. Our previous work [14] revealed that the tensile strength of regenerated cellulose fibers can be enhanced by the addition of nanomaterials, and such improvement was closely related to both the hydrogen bond network within the cellulose solution and the regenerated fiber structure.

The addition of hydrogen bond acceptors (HBA) and hydrogen bond donors (HBD) into cellulose based materials could alter the hydrogen bond distribution, resulting in the enhancement of physical strength. Jiang et al. [15] added urea and other HBAs to the cellulose/AMIMCl solution to provide more interaction sites for the hydroxyl groups of the cellulose. Similarly, as a typical HBD, glycerol was added into the bacterial cellulose/pullulan film, which can bring about a significant increment of both the Young’s modulus and the tensile strength of bio-nanocomposites [16]. When the glycerol was added into the cellulose diacetate films, the hydrogen bonds between glycerol and acetate fibers were generated [17]. Meanwhile, as a typical HBD in the deep eutectic solvents, choline chloride also exhibited good solubility of cellulose [18,19,20]. Furthermore, ILs mixed with co-solvents were also advantageous to increasing their solubility of cellulose [12], indicating that the cellulose dissolution was assisted by the improved hydrogen bond effectiveness of the solution. However, the influences of HBA/HBD addition on the final regenerated cellulose fibers during the cellulose dissolution process are usually neglected.

In this study, the cellulose dissolution and regeneration were achieved by a dry-jet wet-spin approach. Hardwood-dissolving pulp was added into AMIMCl with or without the addition of typical HBAs (choline chloride) and HBDs (urea and glycerol). This process was followed by an investigation of physical strength and thermal stability, and the morphology of the as-prepared regenerated fibers was systematically investigated. The dissolution performance and rheological behavior of cellulose viscose were also evaluated.

## 2. Materials and Methods

### 2.1. Materials

The bleached hardwood-dissolving pulp (DP~548, α-cellulose~88.74%, element content: C~41.49 %, H~6.08%, O~50.78%) was obtained from a pulp mill in Jining, China. N-methylimidazole (Analytical Reagent (AR), >98%) and 3-allyl chloride (AR, >99%) were purchased from Macklin Inc Company (Shanghai, China), and were distilled before use. Urea (AR, >99%), glycerol (AR, >99.5%) and choline chloride (AR, >98%) were purchased from Macklin Inc., Company (Shanghai, China) and were used without further purification.

The 1-allyl-3-methylimidazolium chloride (AMIMCl) was synthesized according to a method reported in the literature [14]. The purity of ~98% was determined by nuclear magnetic resonance spectroscopy (AVANCEII400, Bruke Inc., Billerica, Germany) ((400 MHz, DMSO-d6), 9.882 (s, 1H, NCHN), 8.06~8.08 (d, 2H, NCHCHN), 6.03 (ddt, 1H, CH_2_CH_1_), 5.23 (m, 2H, CH_2_CH), 5.005 (d, 2H, CH_2_N), 3.934 (s, 3H, NCH_3_)) and a water content of ~0.51% was verified by a Karl Fischer moisture meter (AKF-1, Mettler Toledo International Co., LTD, Zurich, Switzerland).

### 2.2. Molecular Simulation

Molecular simulation was conducted with the conductor-like screening model for a real solvents (COSMO-RS) model (BIOVIA COSMOtherm 2020, Version 20.0.0, Dassault Systèmes, Paris, France), in which quantum chemical calculations were combined with statistical mechanics to explore the thermodynamical behaviors of the chemicals used in this work.

The structure of cellobiose was drawn by Turbomole (BIOVIA TmoleX 2021, Version 21.0.0, Dassault Systèmes, Paris, France). The geometry optimizations were performed at the density functional theory (DFT) level and utilized the BP function with resolution of identity (RI) approximation and a triple-ξ valence polarized basis set (TZVP) were utilized [21,22]. All the other chemicals were obtained from the built-in database COSMObaseIL-1901. For the COSMO-RS calculations, it was assumed that ILs were set as a mixture of an equimolar composition of cations and anions.

### 2.3. Preparation of Regenerated Cellulose Fibers

Urea, glycerol, and choline chloride were added into 100 g AMIMCl, and the corresponding contents are given in Table 1. The mixture was stirred and heated at a temperature of 100 °C to form a uniform solution and then cooled down to 70 °C. Subsequently, 10 g bleached hard wood dissolving pulp was added into the above mixture to generate the homogeneous mixture of cellulose-IL.

Figure 1 shows the synthesis procedure of regenerated cellulose fiber. It can be seen that the cellulose fiber was generated by a single screw extruder followed by deionized water as a coagulation bath. Details of the operating condition have been documented in our previous work [14]. The aspect ratio of the extruder was 25:1 and the material extrusion quantity was 2~5 kg per hour. According to the temperature varied from the feeding port to the extrusion head, four temperature zones were set at 150 °C, 165 °C, 120 °C and 80 °C, respectively. The rotor speed was 50 revolutions per minute and the feed rate of the homogeneous mixture was 3 kg per hour. The cellulose solution was extruded through an extruder with a diameter of 0.8 mm, and then passed through 2 cm of air into a deionized water coagulation bath. The wet cellulose fiber was cleaned in a water bath (50 cm × 30 cm × 30 cm) at a speed of 1 m/min and manually winded in a 10 cm diameter glass tube with a stretch ratio of 1. The regenerated cellulose fiber was soaked in deionized water for 24 h and then washed at 80 °C, and then the solvent with room-temperature deionized water was removed until the content of AMIMCl in the regenerated fiber was less than 0.3 wt%. The fiber was then washed with deionized water three times and soaked in 60 g/L glycerol aqueous solution for 3 s. Subsequently, the fibers were dried under 120 °C hot air for 10 min and placed in a balanced temperature and humidity environment for 24 h. The stretch ratio during washing and drying was 1:1, with the fiber manually winded in a glass tube with a diameter of 10 cm diameter. The dosages of chemicals used for regenerated cellulose fiber manufacture are summarized in Table 1.

### 2.4. Characterization of Regenerated Fibers

#### 2.4.1. The Mechanical Properties of the Regenerated Cellulose Fibers

The mechanical properties of the regenerated cellulose fibers were measured by a texture analyzer (stable microsystems PL/CEL5, Stable Micro Systems Inc., London, UK) with a 5000 g load cell. The initial tensile distance was kept at 20 mm, and the tensile rate was 10 mm/min [23,24].

#### 2.4.2. X-ray Diffraction (XRD) Analysis

The X-ray Diffraction (XRD) data were collected on a model D8 ADVANCE diffractometer (Bruker AXS Co., Karlsruhe, Germany) in an angular region 2θ ranging from 5° to 60° with a scan speed of 5°/min. Fibers were spiral wound 20 times on the test sheet and compacted to achieve a complete cover on the test sheet. As demonstrated in Equation (1), the crystallinity can be estimated by the analysis of X-ray diffraction patterns.
(1)$$\mathrm{Crystallinity}=\frac{A_{crystal}}{A_{crystal}+A_{amorphous}}\times 100\%,$$
where *A_crystal_* and *A_amorphous_* were calculated according to the literature method [25,26], using Gaussian fitting. For cellulose II, *A_crystal_* denotes the integral area of the (200) characteristic diffraction peak located at 20.4°, and *A_amorphous_* represents the integral area of the predominantly amorphous peak located at 18.0°.

#### 2.4.3. Scanning Electron Microscopy (SEM)

The scanning electron microscopy (SEM) observation of regeneration cellulose fiber samples was conducted on a regulus 8220 microscope (Hitachi Ltd., Tokyo, Japan) operating at 5 kV with the usage of filtered Cu Kα radiation. The fracture surface of the fibers was coated with a platinum layer (20 nm thickness) before the observation.

#### 2.4.4. Fourier-Transform Infrared (FT-IR) Spectroscopy

The Fourier-transform infrared (FT-IR) spectra of the regenerated cellulose fibers were recorded by an ALPHA FTIR spectrophotometer (Bruker Corporation, Billerica, MA, USA) with attenuated total reflectance (ATR) and diamond-ZnSe crystal. The wavelength range was from 500 to 4000 cm^−1^ with a resolution of 4 cm^−1^. At least three repetitions per sample were conducted. All of the spectra were corrected against an air background. The spectra of all samples were corrected by water vapor, automatic baseline correction, and smoothing by MONIC 9.0 software, and the spectra are presented without ATR correction [27]. The crystallinity index obtained by FT-IR was denoted as CI(IR), which was calculated on the basis of the literature method [28]. The ratios of the band at A_(1427, 1425)_/A_(913, 897)_ and A_(1368)_/A_(1262, 1260)_ were used, in which the band at 1422 (1425), 913 (897), 1368 and 1262 (1260) cm^−1^ was selected as a reference. The total crystallinity index (TCI) was calculated through the ratio of the peaks A_1372_/ A_2900_ (the two peaks are low susceptibility to water) [29].

#### 2.4.5. Thermal Analysis

Thermographic analysis-differential scanning calorimetry (TG-DSC) of the fibers was performed on a DSC analyzer (NETZSCH STA 449F3, Bavaria, Germany). Ten milligrams of the sample were placed in an aluminum oxide pan and heated from 30 to 800 °C with a heating rate of 10 °C/min in nitrogen atmosphere with a flush rate of 40 mL/min. The temperature of beginning of weight loss, maximum rate of weight loss and glass transition was obtained. The glass transition temperature was calculated according to the literature method [30].

#### 2.4.6. Dissolution Analysis

Dissolution analysis was carried out according to the literature method [31,32], with a polarizing microscope (PM6000, Leica Microsystems, Wetzlar, Germany) at 400× magnifications. Briefly, 10 g of pulp was added into 100 g AMIMCl mixtures in a beaker at 120 °C, and the status of cellulose dissolution was recorded at 0,10, 20, and 30 min, respectively.

#### 2.4.7. Rheological Measurements

The cellulose solution samples used for rheological measurements were obtained with a temperature of 80 °C and that of coagulation at 25 °C. Thereafter, the viscosity, stress, and viscoelasticity of the cellulose solution sample were measured at 80 °C with an ARES-G2 rotated rheometer (TA Instruments, New Castle, DE, USA). A parallel plate (25 mm in diameter) with a given gap of 1 mm was used for all procedures. The linear viscoelastic regime of the samples was tested for the analysis of storage modulus and loss modulus. The variations in dynamic viscoelasticity against time, angular frequency, and temperature were analyzed in detail.

## 3. Results

### 3.1. The Sigma Profiles Analyze

Figure 2 illustrates the sigma profiles of the chemicals used in this work. σ values lower than −0.01 e/Å^2^ were the HBD region, σ values between −0.01 e/Å^2^ and 0.01 e/Å^2^ reflected the non-polar region, and values higher than 0.01 e/Å^2^ were the HBA region [33,34]. Water is considered to be both HBA and HBD, and two characteristic peaks located at −0.016 and 0.017 e/Å^2^ can be detected. The cellulose framework was molded using cellobiose as a model compound to reduce calculation loading. Considering the presence of numerous free -OH groups, cellulose with both strong HBD (−0.017 e/Å^2^) and HBA (0.016 e/Å^2^) are noted. Although AMIMCl could dissolve cellulose up to 25% [35], the system is mainly associated with its anions for the dissolution. It suggests that AMIMCl can break the hydrogen bond network of cellulose with its Cl anion and behave as a strong HBA. The modeling results have verified this theory in which a peak lying at 0.019 e/Å^2^ with high intensity is identified for the Cl anion; whereas, the AMIM cation is located in the non-polar region. Importantly, cellulose acts concurrently as both an HBA and HBD during the formation of the hydrogen bond network. By contrast, this system may have presented higher activity as an HBA (higher at 0.016 e/Å^2^ than that obtained at −0.017 e/Å^2^). Therefore, the modification of AMIMCl, particularly with the addition of HBDs, may bring about better cellulose dissolution.

HBDs (urea and glycerol) and HBAs (choline chloride) were typically added to cellulose-ILs mixtures to enhance hydrogen bonding activity in the system. Urea presents a typical HBD peak at −0.015 e/Å^2^ and glycerol had an HBD peak at −0.017 e/Å^2^ (Figure 2). Choline chloride exhibits a typical HBA peak at 0.015 e/Å^2^ for hydroxide groups in choline cation and a peak at 0.019 e/Å^2^ for the Cl anion. It implies that the addition of these HBAs and HBDs has altered the hydrogen bond distribution in the mixture system. Accordingly, the cellulose dissolution can be enhanced and thereby the corresponding physical and chemical properties of regenerated cellulose are improved.

### 3.2. Tensile Properties and XRD Analysis

Regenerated cellulose fibers were obtained with or without the addition of HBA and HBD. As shown in Figure 1 and Figure 3, clear and uniform mixtures can be observed for all samples, indicating that successful cellulose dissolution is achieved for each testing sample. In terms of the regenerated cellulose fibers, the XRD results in Figure 3 demonstrate that the softwood dissolving pulp (raw material in this experiment) is natural cellulose and has the configuration of cellulose I, diffraction peaks at 2θ of 16.1°, 22.2° and 34.5° are indexed as crystal planes of (110), (200) and (004) of cellulose I, respectively [36]. As compared to those reported in the literature, the regenerated cellulose fibers were all different. Some researchers considered that such a crystalline form of the regenerated fibers typically had main diffraction peaks at 2θ of 20.4°, which represented the crystal plane (110) of the cellulose II crystalline form [37,38]. In particular, the width of such a characteristic diffraction peak is larger in present case, suggesting that the that amorphous zone and crystalline zone would coexist [39]. On the other hand, some researchers believed that these XRD patterns would provide the amorphous structure of the regenerated cellulose fiber [26]. In this study, the structure and crystallinity of regenerated cellulose fibers are regarded as cellulose II crystalline. The change in crystallography indicates the cleavage of initial intermolecular hydrogen bonds in cellulose in the AMIMCl solutions, ultimately leading to the dissolution of cellulose as a viscose [40]. As mentioned above, the dissolution and regeneration of cellulose in this work can be described in Figure 1. The cellulose was dissolved in AMIMCl and pressed through the extruder into a water coagulating bath where AMIMCl was rapidly dispersed in the aqueous solution. The cellulose is coagulated with the reformation of hydrogen bond networks. During regeneration, some glucose units of cellulose are accumulated together with secondary helical structures to form a cellulose II configuration [41,42]. Regenerated cellulose II has a more complex structure than cellulose I. A shorter length of hydrogen bonds can induce the closer packing of cell units, while a more staggered hydrogen bond in cellulose II is conductive to obtaining higher physical strength [43,44].

In comparison with the control test (BF sample), the addition of urea, glycerol, and choline chloride causes all the samples to exhibit a higher tensile strength, but with a limited change in crystallinity. The physical strength of regenerated cellulose fiber is usually correlated to the extent of crystallinity, since the tenacity of the regenerated fibers mainly depends on the orientation of their amorphous region, whereas the modulus and crystallinity are related to crystal orientation [45]. The tensile strength and elongation at the break of BF samples are determined as 54.43 MPa and 46.8% respectively. The crystallinity of the BF sample is around 60.06%. With only a ~2 % (molar ratio for AMIMCl) addition of urea (1.89 g in 100 g AMIMCl), the crystallinity reaches 61.58%, and the tensile strength of UF samples increased to 88.57 MPa, by a factor of 67.21%. With regard to the GF sample, as the addition of glycerol is approximately 2.9 g in 100 g AMIMCl, the tensile strength is increased to 105.67 MPa while its crystallinity remained nearly the same as the BF sample. Similarly, the addition of choline chloride (4.40 g in 100 g AMIMCl) as HBA causes the tensile strength of the CCF sample to increase to 67.29 MPa and the corresponding crystallinity reaches 62.13%. It is apparent that adding either HBA or HBD into the AMIMCl would enhance the physical strength of regenerated cellulose fibers, as the initial modulus of regenerated cellulose fibers shows the same trend (Table 2).

The hydrogen bond interactions within the system can be improved by the addition of HBA and HBD. Similarly, the CCUF sample upon the addition of urea and choline chloride leads to a superior physical strength with the tensile strength of 117.36 MPa. Meanwhile, for the CCGF sample, upon the addition of glycerol and choline chloride, the tensile strength is increased to 139.62 MPa with the crystallinity of 62.79%. All these samples have the same molar dosage of additives—either HBA or HBD on their own, or in combination. The CCGF sample and CCUF sample present higher physical strength than the UF, GF, and CCF samples. It manifests that the synergistic effects can be achieved by the addition of HBA and HBD together into AMIMCl, and thus the mechanical strength of regenerated cellulose fibers is dramatically improved.

Besides, the glycerol-treated samples possess a prominent enhancement in physical strength but their crystallinity is maintained as constant. In comparison with the BF sample, the sample with the addition of urea (UF) and choline chloride (CCF) gained minor elevation in physical strength along with same crystallinity. The crystallinity of the GF sample reaches 61.89%, which is close to those of other samples, but the tensile strength of the regenerated fiber impressively acquired a reinforcement. Similar effects can be noticed when comparing CCGF samples with CCUF and CCF results. Undoubtedly, in the present case, the crystallinity cannot be the main factor affecting the physical strength of regenerated cellulose fibers, while some other factors, including morphology, chemical crosslinking and rheology performance, probably make greater contributions during this process.

### 3.3. SEM Observation

Figure 4 provides the surface morphology of nature pulp and the regenerated fibers; the natural softwood-dissolving pulp (Figure 4a) is featured with a coarse stratiform fiber surface, whereas all the regenerated cellulose fibers possess smooth column structures covered with the compact squamous texture on their surface. This could be attributed to the recrystallization of glucose macromolecules chains in the coagulating bath during cellulose reaggregation. Therefore, the uniformity and surface smoothness of regenerated cellulose fibers are remarkably improved after dissolution and dry-spinning.

Figure 4b sheds light on the fact that the BF sample has a squamous structure on its surface with few irregular pores, protrusions and cracks. These pores can be ascribed as typical characteristics of the regenerated cellulose fibers [46,47] and provide the fibers with good air permeability. It is obvious that the smoothness is significantly improved after the addition of urea. By contrast, the UF sample (Figure 4c) exhibits fewer protrusions and cracks, and has a smaller pore size with a uniform distribution on its surface. Accordingly, after adding urea into the dissolution system, the occurrence of structure evolution can be found in the regenerated fibers, and thereby the related optimal distribution of urea in the coagulation bath can also be obtained.

As indicated in Figure 4d, the GF sample has a fiber surface with enlarged squamous structures and fewer pores, and no cracks could be observed on the surface. Glycerol can produce a more smooth surface of regenerated cellulose [48], and commonly serves as a plasticizer or softener during the manufacture of cellulose films. It is probable that the addition of glycerol in AMIMCl has altered the macromolecular structure of cellulose by increasing the number of oxygen atoms with free electron pairs, thus is favorable for increasing flexibility of the group [49] with hydrogen bonds to the cellulose [50]. The CCF (Figure 4e) and CCUF (Figure 4f) samples exhibit a similar morphology as the CCGF (Figure 4g) sample and are featured with wide squamous structures (Figure 4g). The surfaces of CCUF and CCGF samples are denser, and a small amount of protrusions can be examined.

Therefore, during the cellulose dissolution, the addition of HBA and HBD either on its own or in combination with AMIMCl can bring the change of the surface morphology of regenerated cellulose fibers to increase the strength and promote the aggregation. This phenomenon is consistent with the physical strength results previously presented in this work. The fiber strength enhancement can be attributed to the fact that, during the regeneration process, more compact structures and smoother surfaces can be produced, and the number of protrusions and cracks are reduced. These results, however, cannot explain why the glycerol-treated samples can provide excellent physical strength with only a minor elevation in crystallinity and similar morphology of regenerated cellulose fibers (GF sample and CCGF sample). Thereby, further detailed characterizations of the regenerated fiber and the cellulose-IL solution need to be carried out.

### 3.4. FT-IR Spectroscopy Analysis

Figure 5 illustrates the FT-IR spectroscopy results of regenerated cellulose fibers with or without the addition of HBAs and HBDs. All the regenerated cellulose fibers display similar spectra, suggesting that the basic structure of the cellulose frameworks are well preserved (Figure 5a). Typical cellulose structural features are listed in Table 3. Figure 5b shows the details of the FT-IR spectra with the wavenumber between 3800 and 2500 cm^−1^. The frequency band between 3600 and 3100 cm^−1^ is identified as the O-H stretching bond (3-OH⋯O-5) of hydrogen bonds of cellulose. The presence of the 3600~3100 cm^−1^ band in all testing samples confirms the well preserved cellulose framework. The band between 2940 and 2840 cm^−1^ is ascribed to the characteristics of the -CH and CH_2_ stretching bond of cellulose [51,52]. As compared to the BF sample, with the characteristic peak appearing at 2920 and 2885 cm^−1^, the wavenumber of samples with additives shows a bule-shift of the -CH stretching bond to 2928 cm^−1^ and a red-shift of the -CH_2_ stretching bond to 2861 cm^−1^. This may be derived from the presence of choline chloride and glycerol in the cellulose fibers [53], which could affect the -CH bond (blue shift) and induce the enhancement/formation of hydrogen bonds between the hydroxyl group on C6 and the additives (red shift).

Figure 5c shows the details of the FT-IR spectra with a wavenumber between 1500 and 1150 cm^−1^, where the bands are majorly allocated to the bonds in cellulose II. The bands between 1500 and 1200 cm^−1^ are attributed to the -OH in-plane bending of cellulose II. The band at 1427~1425 cm^−1^ and 1325 cm^−1^ originated from -CH_2_ bending or scissoring motions in both cellulose II and amorphous cellulose [59]. The peaks at 1331~1325, 1164~1156, 1019~985, and 913~897 cm^−1^ belong to the characteristic peaks of OH in-plane bending, deformations of C-O-C group, C-O stretching of internal hydrogen bonds, and β-glycosidic linkages between the sugar units of cellulose, respectively. All these peaks can be assigned to the crystalline phase of cellulose II [29]. The mentioned characteristic peaks related to cellulose II can be found in all the testing samples, which further confirms the successful cellulose dissolution and regeneration in the process. This finding is in line with the XRD results. The -CH_2_ bending bond at 1425 cm^−1^ (BF sample) is slightly shifted to 1427 cm^−1^ in the CCGF sample, the chemical environment of the hydroxymethyl group attached to the C6 in glucose unit is changed by such treatment. It can be deduced that a small amount of additive is involved in the regeneration of cellulose and still remained within the final product, thus affecting the intermolecular bond at C6 in the glucose unit.

Figure 5d illustrates the FT-IR spectra with a wavenumber between 1200~650 cm^−1^. The band at 1019 and 913 cm^−1^ in the BF sample are shifted to 985 and 897 cm^−1^ in the samples with additives. These changes are related to the transformation of intra- and intermolecular bonds [28], which shows stronger/more hydrogen bonds together with a tight arrangement of the cellulose chains. The peak at 1650 (Figure 5a) and 700 cm^−1^ could be attributed to the hydrogen-bound water within the cellulose fiber, indicating that the water is not completely removed and becomes part of the hydrogen bond network within regenerated cellulose [60]. The peak at 1466 cm^−1^ (Figure 5c) is determined to be the characteristic bands of N-H stretching vibration. It is obvious that the peak at 799 cm^−1^ can be identified in GF, CCF, CCUF and CCGF samples. It is speculated that the urea and choline chloride could interact with cellulose and remain in the fiber after regeneration [55].

The crystallinity index (CI_(IR)_) and total crystallinity index (TCI) of regenerated cellulose fibers with or without additives are displayed in Table 4. The peaks at 1425 and 897 cm^−1^, which were related to the amorphous and crystalline structures in the cellulose, are regarded as the characteristics of the C-OH in plane at C6 [28]. According to Table 4, all the testing samples display an intensity ratio of A_(1427, 1425)_/A_(913, 897)_ and A_(1368)_/A_(1262, 1260)_ located within the range of 99%~100%. It suggests that there is no obvious change in the hydrogen bonding and crystal structure of cellulose (C6) in all the testing samples. The TCI values are calculated on the basis of the method reported by Nelson et al. [59], where all the regenerated cellulose fibers, with or without additives, showed similar results. This finding further supports the XRD analysis that all the testing samples are composed of cellulose II and amorphous cellulose. Thereby, both the XRD and FT-IR results demonstrate that the crystal phase and crystallinity of the regenerated cellulose fibers are basically unchanged with or without additives.

Therefore, it is clear that the cellulose framework can be well preserved in the dissolution and regeneration with or without the addition of HBAs and HBDs. All the regenerated fibers are composed of cellulose II and amorphous cellulose. No sign of chemical reactions can be examined for the samples that has urea, glycerol, and choline chloride added to them. Nevertheless, the FT-IR results have qualitatively identified the presence of HBA/HBD with a trace amount in the regenerated cellulose fibers mostly through hydrogen bonds. These HBA/HBD residues are probably responsible for the cross-linking of cellulose macromolecules, thus enhancing the physical strength of the fibers.

### 3.5. Thermal Analysis and Differential Scanning Calorimetry Analysis

Figure 6 displays that all the regenerated cellulose fibers contain thermal degradation in three stages. The initial stage starts within the temperature range from 30 to 150 °C and this is mainly derived from the volatilization of free water and the removal of unstable substances within the fibers. The second stage is located within the temperature range of 150 to 250 °C, and the decomposition of amorphous regions and deep dehydration of glucose are detected. The third stage shows the fastest degradation of cellulose at a temperature between 250 and 400 °C and was caused by the cleavage of glycosidic bonds [61]. As shown in Table 5, the onset decomposition temperature of the BF sample is determined as 162.64 °C, and then the second decomposition phase is generated with a sharp weight loss. Such weight is induced by the degradation of cellulose glucose groups in the regenerated fibers [62,63].The onset temperature of the BF sample is the lowest among all the tested samples. The addition of HBA and HBD can bring about an increment in onset decomposition temperature; the T_on_ for UF, GF, CCF, CCUF, and CCGF samples are 186.70 °C, 177.09 °C, 186.20 °C, 184.80 °C, and 182.89 °C, respectively, indicating that the addition of HBA/HBD is conducive to the regenerated cellulose fibers with a higher onset decomposition temperature. Meanwhile, the maximum rates of weight loss temperatures of all the samples are almost the same. Therefore, the addition of HBA/HBD causes only limited changes in the thermal stability of regenerated cellulose fibers.

As shown in Table 5, the TG-DSC results indicate that the Tg of fibers is improved by the addition of glycerol, and the treatment with urea and choline chloride can produce the same positive effects as glycerol. The Tg values of the fibers are also promoted by the addition of both HBA and HBD. The Tg of GF and CCGF samples are increased to 152.32 °C and 178.66 °C, respectively, suggesting that these additives are located within the crystalline region and the amorphous region. The additives could combine with each other and interact with cellulose molecules through hydrogen bonds in their amorphous region. With the assistance of FT-IR results, these HBA/HBD residues could provide a great contribution to the closer bonding between molecules in the amorphous region, leading to an enhancement in the thermal stability of the regenerated cellulose fibers.

### 3.6. Cellulose Dissolution Analysis

A polarizing microscope was applied to observe the dissolution of cellulose [31,32], where successful cellulose dissolution should present a vision field without a bright spot. As shown in Figure 7, the reinforced dissolution of cellulose can be found in the UF and CCF samples versus the BF sample. The UF and CCF samples are efficiently dissolved within 20 min, and only a few fragments could be viewed at 30 min. This finding suggests that the addition of urea and choline chloride improves the hydrogen bonding integration within the system, thereby facilitating a disturbed cellulose H-bond network to ultimately promote the occurrence of dissolution. In the CCF sample, the amount of hydrogen bond acceptors in the system is increased by the addition of choline chloride, which was reported to improve the solubility of cellulose in an IL system to accelerate dissolution [64]. Similar effects are also found in the urea-treated sample. Interestingly, the dissolution rate of the CCUF sample is lower than those of the UF and CCF samples, since more cellulose fibers can be observed at 30 min. When the urea and choline chloride are concurrently added, the dissolution rate is not synergistically accelerated. It can be speculated that urea (HBD) and choline chloride (HBA) could be combined with each other, and thus the probability of a hydrogen bond binding with cellulose is reduced. In terms of GF and CCGF samples, it seems that adding glycerol could protect the fibers from dissolution, when the GF and CCGF samples are compared with BF and CCF samples, respectively. It can be seen from Figure 7 that there is lack of obvious damages within GF and CCGF samples at 10 min, and well-preserved cellulose fibers could be found at 30 min. The unusual behavior in cellulose dissolution may produce these unique rheological properties in the glycerol-treated samples, leading to an excellent physical strength [65].

### 3.7. Rheological Properties

The rheological properties of cellulose-AMIMCl solution are exhibited in Figure 8. All the samples are shear-thinning fluids with a yield stress value of 80 °C (Figure 8a). Notably, their non-Newtonian fluid characteristics remain unchanged after the addition of urea, glycerol, and choline chloride. The viscosity of the cellulose solution is changed slowly at a low shear rate; the yield stress and the viscosity values are followed by the sequences of UF > CCUF > GF > CCGF > BF ≥ CCF. Figure 8b shows that, as the share rate is increased from 0.01 to 10 1/s, the stress of cellulose solution follows a straight-line trend. The viscosity and stress of cellulose solution are increased after the addition of HBA and HBD, but the solution with CCF is exceptional. New hydrogen bonds are formed between the urea, glycerol, choline chloride, AMIMCl, and cellulose within the UF, CCUF, GF, and CCGF samples. The macromolecules can induce the formation of an H-bond networked structure in the viscose. The viscosity of these samples is dramatically increased, indicating that the strength of the polymer materials is also improved [66]. As expected, the increased viscosity of cellulose-AMIMCl solution can promote the enhancement of the physical strength of regenerated fibers.

All the cellulose solution samples reveal a typical shear thinning behavior at a high shear rate, especially above a shear rate of 10 1/s. Cellulose and ILs could construct a macromolecular chain containing a hydrogen bond network [15], which could be orientated with increasing shear rate, leading to a reduction in viscosity. The increased shear rate and the resultant shear force could break the entanglement and intermolecular binding of cellulose-ILs macromolecules, and then the dynamic equilibrium of the viscose samples are further destroyed. Therefore, the occurrence of physical cross-linking and partial remodeling of hydrogen bonds, as well as the relative slip and rearrangement, can be found between cellulose-ILs macromolecules.

Figure 8c shows the storage modulus (G’) and loss modulus (G”) of the all cellulose-IL solution at 80 °C within the linear viscoelastic regime. The G” of the BF sample is always larger than G’ within the range of oscillation frequency at 0–100 rad/s. This result unveiled that the BF sample has a flow dynamic state, while the G’ and G” are decreased upon the addition of HBD and HBA, and the viscosity and elasticity of cellulose solution are enhanced. At low angular frequency, their G’ values are larger than G”. However, at high angular frequency, the increase of G” value is more dramatic than that of G’. Additionally, the appearance of crossover points can be found in Figure 8c. The variation tendencies of UF, GF, CCF, CCUF, and CCGF samples are totally different. Accordingly, this crossover point is the gel point of the cellulose solution [67,68]. Their structure could be rebuilt herein, which indicated the transformation of the entanglement of polymer molecules. Thus, owing to the addition of HBA and HBD, the hydrogen bonds between the components in the cellulose solution become more complex and tighter.

From these above results, it is obvious that the addition of HBA and HBD can change the rheological properties of cellulose-AMIMCl viscose, and then increase the viscosity and storage modulus and make a gel point to the cellulose solution. All these changes are responsible for the improved physical strength of the fibers.

## 4. Conclusions

In this study, cellulose dissolution and regeneration were achieved through dry-jet and wet-spinning. As suggested by COSMO-RS modeling results, the 1-allyl-3-methylimidazolium chloride (AMIMCl) worked only as HBAs while dissolving the cellulose, and this process can be enhanced by additives. The hydrogen bonding ability of the AMIMCl system was adjusted by the addition of urea, glycerol, and choline chloride either on their own or in combination. The addition of HBDs and HBAs can bring about variations in the tensile strength, the crystallinity and the thermal stability of the regenerated cellulose fibers. A more compact structure and fewer pores on the fiber surface were identified in the sample with additives along with a well-preserved cellulose framework. The amorphous region structure of cellulose macromolecules was deeply disturbed by a small amount of residual additives in regenerated cellulose fiber, leading to the occurrence of cross-linking between cellulose macromolecules.

Interestingly, the tensile strength was dramatically increased by the addition of glycerol (2% molar ratio) in AMIMCl systems, whereas their effects on crystallinity are quite limited. As compared to the additive-free samples, the sample with the addition of glycerol presented an increase of 94% and only 3% in tensile strength and crystallinity respectively. In order to realize further optimization, the glycerin and choline chloride were added simultaneously, and then the tensile strength and crystallinity were increased by 156.50% and only 4.54%, respectively. The remarkable effect of glycerol was attributed to reducing the damage of cellulose by slowing down the dissolution and cross-linking in the cellulose solution. Accordingly, it is feasible to improve the physical strength of the regenerated cellulose fibers by properly adjusting the hydrogen bond distribution in IL systems. The concurrent addition of glycerol and choline chloride can be favorable for achieving regenerated cellulose fibers with outstanding properties.

## Figures and Tables

**Figure 1 polymers-14-02030-f001:**
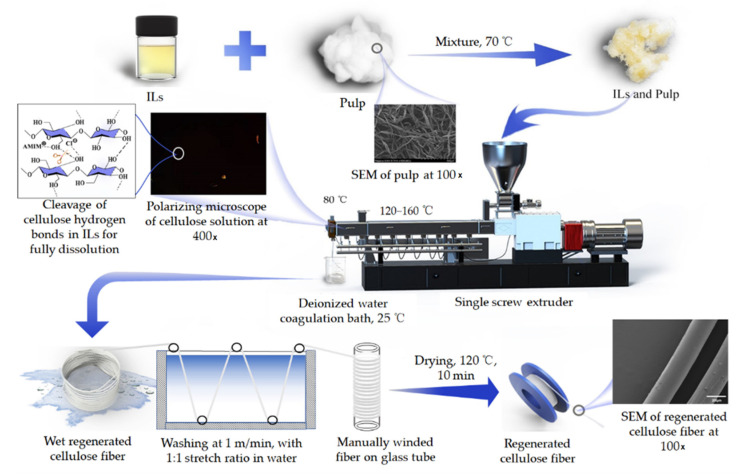
The synthesis procedure of regenerated cellulose fiber: The ionic liquids (ILs) and pulp mixture were put into a single screw extruder, where the cellulose was fully dissolved to form a homogeneous solution (polarizing microscope at 400X, with limited bright spot). The cellulose was regenerated in the coagulation bath, followed by washing and drying to obtain the regenerated cellulose fiber.

**Figure 2 polymers-14-02030-f002:**
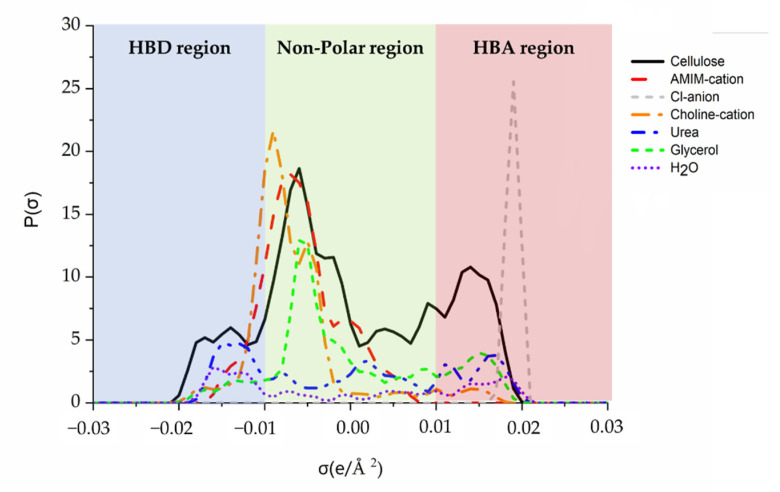
Sigma profiles of cellulose, AMIMCl, urea, glycerol, and choline chloride generated by COSMO.

**Figure 3 polymers-14-02030-f003:**
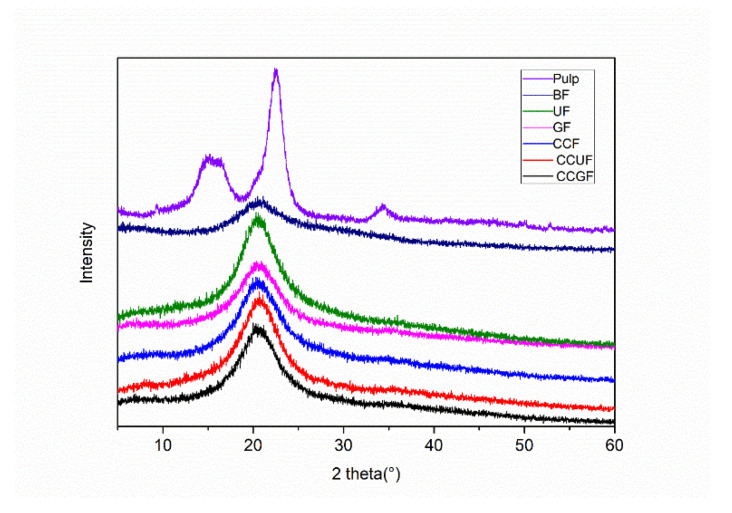
X-ray diffraction patterns of pulp and regenerated cellulose fibers with or without additives.

**Figure 4 polymers-14-02030-f004:**
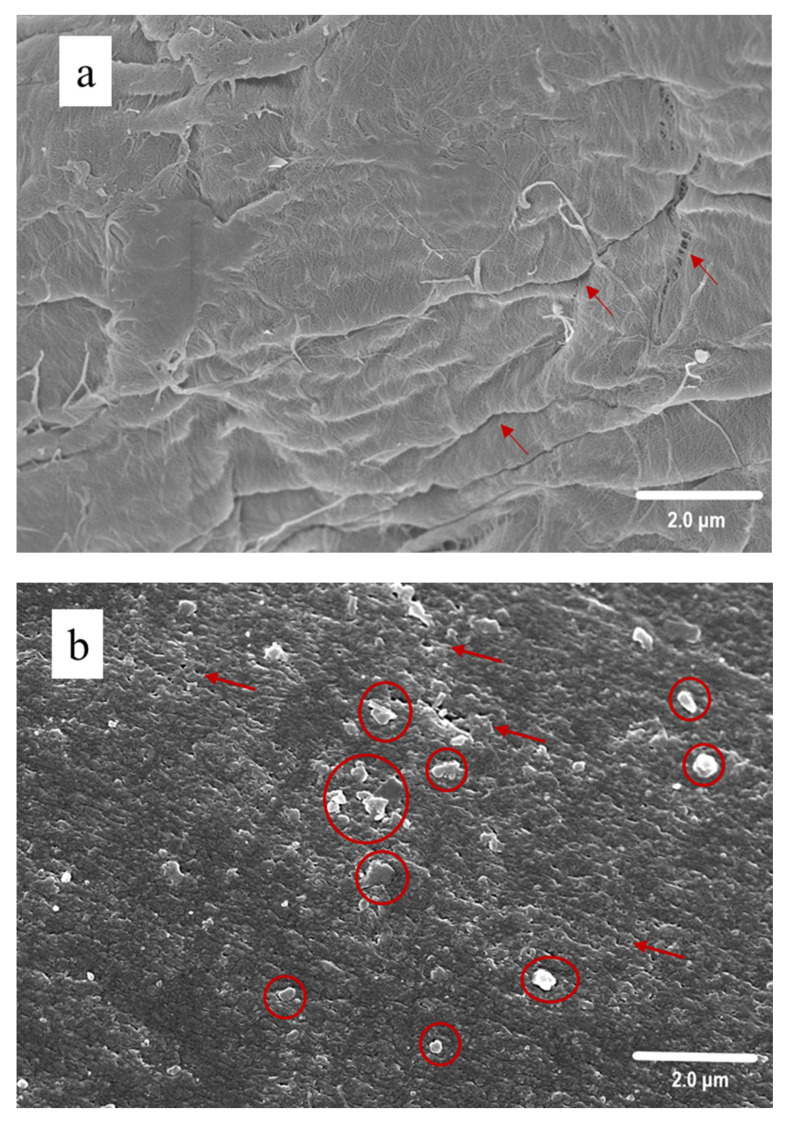
SEM Images of the surface of the pulp and regenerated cellulose fibers: Pulp (**a**), BF (**b**), UF (**c**), GF (**d**), CCF (**e**), CCUF (**f**), and CCGF (**g**). The irregular pores and cracks (marked by arrows), and the protrusions (marked by cycles) are given in red highlight.

**Figure 5 polymers-14-02030-f005:**
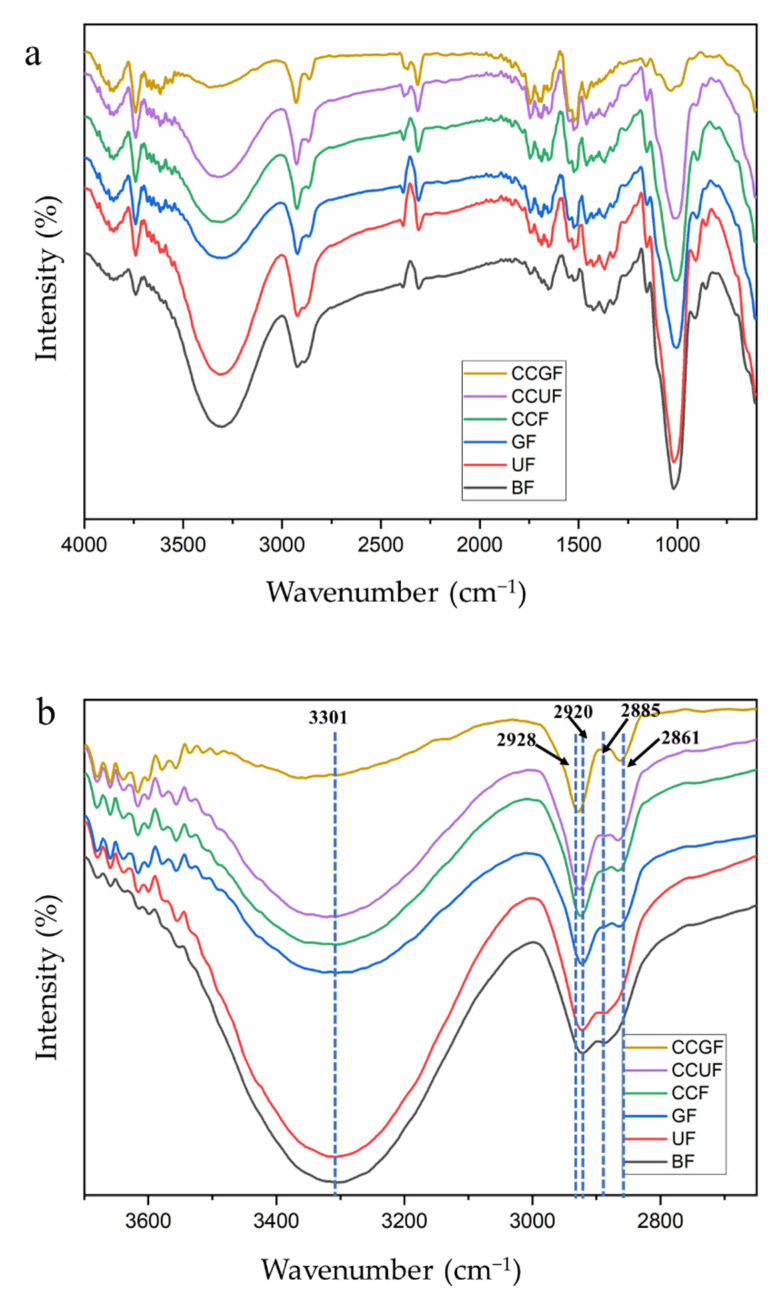
FT-IR spectra of the regenerated cellulose fibers with or without additives: (**a**) FT-IR spectra of the wavenumber at 4000~600 cm^−1^; (**b**) FT-IR spectra of the wavenumber at 3800~2650 cm^−1^; (**c**) FT-IR spectra of the wavenumber at 1500~1150 cm^−1^; (**d**) FT-IR spectra of the wavenumber at 1200~650 cm^−1^.

**Figure 6 polymers-14-02030-f006:**
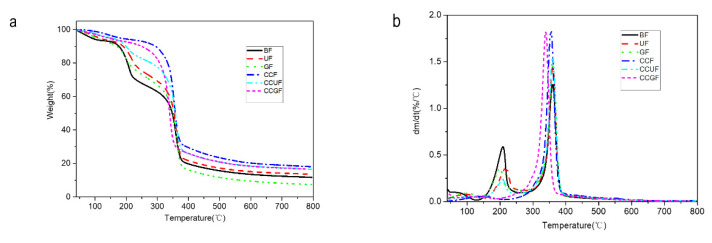
TGA curves (**a**) and DTG curves (**b**) of the regenerated cellulose fibers with or without additives.

**Figure 7 polymers-14-02030-f007:**
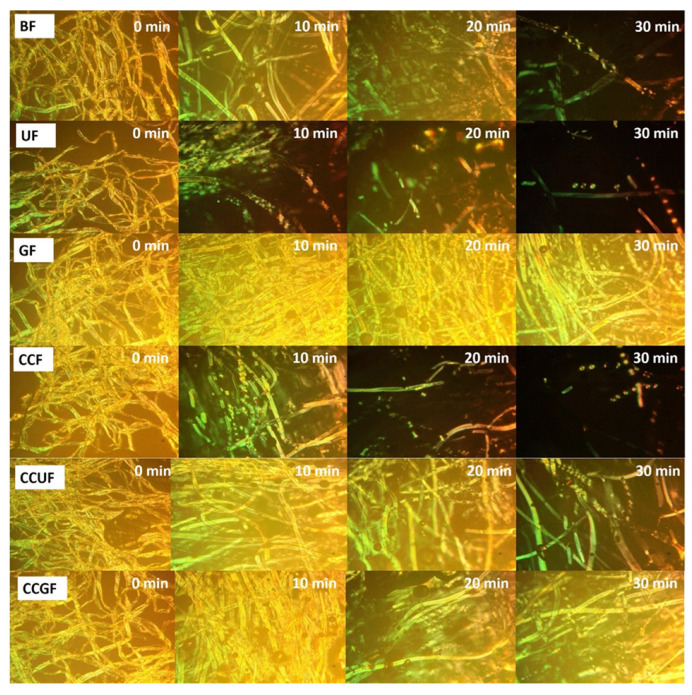
Polarizing microscope observations of the cellulose solubility in AMIMCl solution at 80 °C with 400X magnification with or without additives at 0, 10, 20, and 30 min respectively: The undissolved cellulose appeared as a bright spot in the visual field, and the dissolved cellulose cannot be observed (black).

**Figure 8 polymers-14-02030-f008:**
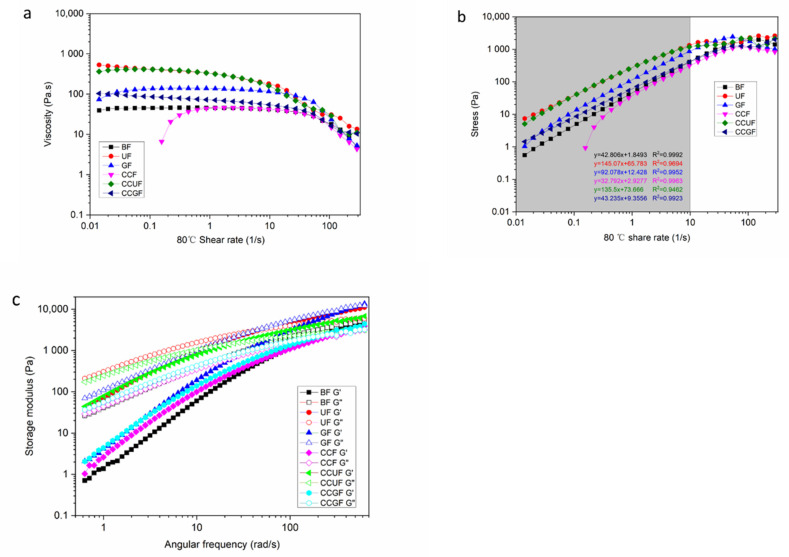
The viscosity (**a**) and shear stress (**b**) of cellulose solution systems with different dissolution behaviors, angular frequency on storage modulus (G’) and loss modulus (G”) of cellulose-IL viscose systems at 80 °C (**c**).

**Table 1 polymers-14-02030-t001:** Dosages of chemicals used in the regenerated cellulose fiber-making procedure.

Sample	Named	AMIMCl (g)	Pulp (g)	Urea (g)	Glycerol (g)	Choline Chloride (g)
AMIMCl	BF	100	10	/	/	/
AMIMCl+ Urea	UF	100	10	1.89	/	/
AMIMCl+ Glycerol	GF	100	10	/	2.9	/
AMIMCl+ Choline Chloride	CCF	100	10	/	/	4.4
AMIMCl+ Urea+ Choline Chloride	CCUF	100	10	1.26	/	1.46
AMIMCl+ Glycerol + Choline Chloride	CCGF	100	10	/	1.16	2.64

**Table 2 polymers-14-02030-t002:** Tensile strength and crystallinity of regenerated cellulose fibers with or without additives.

Sample	Tensile Strength (MPa)	Initial Modulus (MPa)	Elongation at Break (%)	Crystallinity (%)
BF	54.43 ± 2.74	382.33 ± 10.66	46.8 ± 5.32	60.06 ± 2.11
UF	88.57 ± 3.54	815.89 ± 9.57	33.24 ± 1.79	61.58 ± 2.36
GF	105.67 ± 4.50	2314.83 ± 26.56	13.34 ± 4.38	61.89 ± 1.76
CCF	67.29 ± 2.35	1323.40 ± 14.89	14.35 ± 3.59	62.13 ± 1.44
CCUF	117.36 ± 4.28	3166.16 ± 30.22	13.02 ± 4.31	62.50 ± 1.02
CCGF	139.62 ± 5.94	3611.72 ± 27.45	14.78 ± 2.54	62.79 ± 2.00

**Table 3 polymers-14-02030-t003:** Assignment of FT-IR spectra of the regenerated cellulose fibers with or without additives.

Wavenumber (cm^−1^)	Bond and Functional Group	Reference
3600~3100	O-H stretching bond of hydrogen bonds of cellulose	[29,54,55,56]
2940~2840	-CH stretching bond of cellulose	[50,57]
1650	H-O-H bending of the water	[42,50]
1500~1200	-OH in plane bending of cellulose II	[57]
1430	-CH_2_ bending or scissoring motions in cellulose	[29,47]
1315~1338	O-H in-plane bending of the crystalline of cellulose II	[29]
1135~1180	deformations C-O-C groups of cellulose	[42,55]
1050	C-O stretching of internal hydrogen bonds between cellulose	[58]
897	β-glycosidic linkages between the sugar units of cellulose II	[42,57]

**Table 4 polymers-14-02030-t004:** The crystallinity index (CI_(IR)_) and total crystallinity index TCI of regenerated cellulose fibers with or without additives from the FT-IR spectra.

Sample	CI_(IR)_ A _(1427, 1425)_/A _(913, 897)_	CI_(IR)_ A _(1368)_/A _(1262, 1260)_	TCI A _(1368)_/A _(2920)_
BF	99.94%	99.04%	101.47%
UF	99.74%	98.81%	101.27%
GF	100.05%	99.18%	100.97%
CCF	100.07%	99.19%	101.23%
CCUF	99.69%	99.23%	101.23%
CCGF	99.24%	99.59%	100.52%

**Table 5 polymers-14-02030-t005:** Thermal properties of the regenerated cellulose fibers after different stage of treatment.

Sample	T_on_ ^1^ (°C)	T_max_ ^2^ (°C)	T_g_ ^3^ (°C)
BF	162.64	359.86	148.22
UF	186.70	362.76	149.21
GF	177.09	362.72	152.32
CCF	186.20	357.49	147.69
CCUF	184.80	361.62	157.26
CCGF	182.89	358.49	178.66

^1^ T_on_ represents the temperature corresponding to the beginning of weight loss. ^2^ T_max_ represents the temperature corresponding to the maximum rate of weight loss. ^3^ Tg represents the glass transition temperature.

## Data Availability

The data presented in this study are available on request from the corresponding author.

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
