# Peer review of "Strength Enhancement of Regenerated Cellulose Fibers by Adjustment of Hydrogen Bond Distribution in Ionic Liquid"

_polymers, 2022, doi:10.3390/polym14102030_

Round 1

Reviewer 1 Report

The paper “Strength Enhancement of Regenerated Cellulose Fibers by Adjustment of Hydrogen Bond Distribution in Ionic Liquid” reports on the regeneration of cellulose fibers in the presence of ionic liquids and some additives: glycerol, urea and choline chloride. The evaluation of the physical strength in the regenerated cellulose fibers was estimated by mechanical, X-Ray diffraction analysis, thermal, rheological, dissolution tests and SEM microscopy.

While the work seems to be well performed, there are some minor problems with the presentation of the work in this manuscript.

I have the following comments and suggestion:

-Abstract: The authors should remove the paragraphs: “A small number of additives remaining in the regenerated cellulose fibers disturbed the amorphous region structure of cellulose macromolecules, leading to cross-linking between cellulose macromolecules.”- this is a simple assumption, not a proved result.

“ The enhancement of physical strength of regenerated cellulose fiber can be realized by adjustment of the hydrogen bond distribution in the ionic liquid system with additives, and the addition of glycerol and choline chloride together into AMIMCl was preferred. –this was previously mentioned in the first two sentences in Abstract

Introduction

There is no mention about the motivation of using glycerol, urea and choline chloride as additives. Some literature reports should be added in addition to their effects on the structural feature of cellulose derivatives.

Materials and Methods

In Figure 1, there are some typos. Also the figure of the viscose at 400 magnification is not clear, also the chemical structure of cellulose should be more clear. Even if the experimental designe was previously reported in ref. [13], the authors should add some explanations as the legend of the figure on every step.

In table 1 is missing the cellulose’ amount, there are only the additives percents, even if the Table should present the composition of the regenerated cellulose fibers.

The dissolution analysis was estimated only by microscopy? Sounds like a simple assumption to me. Please improve with supplementary experimental data to support the idea.

I appreciate that the authors study the HBD-HBA character of the compounds. Based on these data, the authors should improve the dissolution discussion.

The crystallinity indexes should be also correlated with those calculated from infrared spectra. Based on these data it would be good to discuss the role of the additives on the cellulose fibers strength.

The SEM figures are not clear even if the authors added some comments on them. It would be better to improve them to highlight the differences discussed.

On the infrared spectra there is some confusion. The authors claim that the band at 1050 cm-1 is an absorption maximum. Unfortunately, the spectra are registered on ATR mode, so that there is a baseline point (minimum). So that the discussion on IR spectra should be corrected. Also, the discussion of the infrared crystallinity indexes should be added, based on the authors’ comments.

The thermal analysis does not support the strength enhancement, there are some minor differences. The authors also claim that in Conclusions.

Based on the above observations, I consider that important improvements on this papar should be added before the acceptance. I recommend Major revision.

Author Response

Responses to Reviewer 1 Comments

The paper “Strength Enhancement of Regenerated Cellulose Fibers by Adjustment of Hydrogen Bond Distribution in Ionic Liquid” reports on the regeneration of cellulose fibers in the presence of ionic liquids and some additives: glycerol, urea and choline chloride. The evaluation of the physical strength in the regenerated cellulose fibers was estimated by mechanical, X-Ray diffraction analysis, thermal, rheological, dissolution tests and SEM microscopy.

While the work seems to be well performed, there are some minor problems with the presentation of the work in this manuscript.

I have the following comments and suggestion:

Comment 1:

-Abstract: The authors should remove the paragraphs: “A small number of additives remaining in the regenerated cellulose fibers disturbed the amorphous region structure of cellulose macromolecules, leading to cross-linking between cellulose macromolecules.”- this is a simple assumption, not a proved result.

Response 1:

Thanks for the suggestion. We have removed the mentioned paragraphs in the he revised manuscript. Please check the updated abstract for more details.

Comment 2:

“ The enhancement of physical strength of regenerated cellulose fiber can be realized by adjustment of the hydrogen bond distribution in the ionic liquid system with additives, and the addition of glycerol and choline chloride together into AMIMCl was preferred. –this was previously mentioned in the first two sentences in Abstract

Response 2:

Thanks for your careful check!  We have deleted the duplicate parts in Line 28~29 of the revised paper. Please check the updated abstract for more details.

Comment 3:

Introduction

There is no mention about the motivation of using glycerol, urea and choline chloride as additives. Some literature reports should be added in addition to their effects on the structural feature of cellulose derivatives.

Response 3:

Thanks for the constructive suggestions! We have expanded the introduction to include literature works about the additives used in this manuscript. And to have a better presentation of the key information of the manuscript. Please check the changes with yellow highlight in Line 45-52 and Line 68-80 in Page 2.

Comment 4:

Materials and Methods

In Figure 1, there are some typos. Also the figure of the viscose at 400 magnification is not clear, also the chemical structure of cellulose should be more clear. Even if the experimental designe was previously reported in ref. [13], the authors should add some explanations as the legend of the figure on every step.

Response 4:

Thank you for the comments. The Figure 1 was modified according to the suggestions. We have also expanded the corresponding description in Page 3-4 to show more details of the preparation of regenerated cellulose fibers. Please find the updated Fig 1 in Page 3 and the changes with yellow highlighted in Line 117-120 and Line 131-134 in Page 3-4.  

Comment 5:

In table 1 is missing the cellulose’ amount, there are only the additives percents, even if the Table should present the composition of the regenerated cellulose fibers.

Response 5:

Thanks for your comments. The Table 1 should be the Dosage of chemicals used in the regenerated cellulose fiber-making procedure. Both the title and the content of Table 1 was amended according to the suggestion. Please find the changes in Table 1 with yellow highlight in Page 4.

Comment 6:

The dissolution analysis was estimated only by microscopy? Sounds like a simple assumption to me. Please improve with supplementary experimental data to support the idea.

Response 6:

Sorry for the confusion that was made. The dissolution analysis was designed to show the natural dissolution of cellulose in ionic liquid without vigorous stirring. The experiment was carried out according to the literature method (REF 27 and 28), by using the polarizing microscope. It is an easy and direct method, as the undissolved cellulose appeared as a bright spot in the visual field, and the dissolved cellulose cannot be observed (black). The key finding was reported in Fig.7.

And thanks for the suggestion, we have amended the corresponding paragraphs in Methods to add the related REF (Line 175, P5) and expanded discussions in Results (Line 423-424 and Line 436-437, P15) with more details, please check the revised manuscript for more details.

Comment 7:

I appreciate that the authors study the HBD-HBA character of the compounds. Based on these data, the authors should improve the dissolution discussion.

Response 7:

Thanks for point this out. We have expanded the dissolution discussion in the manuscript, we added more details to explained the effect of HBD/HBA in cellulose dissolution in the ILs. All the changes are highlighted in yellow color, please see the changes in Page 13 of the FTIR results, Page 14 of the TGA results, and Page 15 of the cellulose dissolution results.

Comment 8:

The crystallinity indexes should be also correlated with those calculated from infrared spectra. Based on these data it would be good to discuss the role of the additives on the cellulose fibers strength.

Response 8:

Thank you for the valuable suggestion. We have expanded the discussions in Page 13 and 14, to correlate the FT-IR results with the XRD results. Please find the changes marked with yellow highlight in Line 345-347 and Line 353-358. The role of the additives on the cellulose fibers strength could be find in Line 374-376.

Comment 9:

The SEM figures are not clear even if the authors added some comments on them. It would be better to improve them to highlight the differences discussed.

 Response 9:

Thanks for your comments. We have updated Fig.4 to highlight the irregular pores, protrusions and cracks. The morphology changes caused by the addition of HBA and /or HBD majorly presented different smoothness and the appearance of protrusions and cracks. We also amended the corresponding paragraphs in discussions (Line 308-335, P11-12), please check the revised manuscript for more details.

Comment 10:

On the infrared spectra there is some confusion. The authors claim that the band at 1050 cm-1 is an absorption maximum. Unfortunately, the spectra are registered on ATR mode, so that there is a baseline point (minimum). So that the discussion on IR spectra should be corrected. Also, the discussion of the infrared crystallinity indexes should be added, based on the authors’ comments.

Response 10:

Thank you for point this out. We have amended the paragraphs corresponding to the 1050 peak, meanwhile the discussion combining FTIR and crystallinity was added. Please check Line 359-361 in Page 13 in the revised manuscript for more details.

Comment 11:

The thermal analysis does not support the strength enhancement, there are some minor differences. The authors also claim that in Conclusions.

Response 11:

Thanks for point this out. The main finding here is that regenerated fiber with the addition of HBA/HBD presented higher thermal stability that that of the blank test (BF sample). However, only minor changes could be identified within the samples with additives.

We have updated the TGA discussions (Line 397-415 in Page 14) and also amended the corresponding part in conclusion (Line 495-500 in Page 17). Please find the revised manuscript for more details.

Reviewer 2 Report

Although the authors have conducted a series of expected analytical studies to develop and justify a superior method for spinning fibers from pulp, there is still a lot left to be desired. Even the basic premise that whatever behavior is exhibited by the cellulose/[AMIM]Cl system could be explained away by using a linear HBD and HBA scale is faulty. It is generally known in the field that ionic liquids dissolve cellulose via the formation of electron acceptor AND electron donor complexes and not just the one thing or another (Cao et al., 2016, doi.org/10.1007/s00253-016-8057-8). The premise of this study simplifies the complex dissolution process, and therefore cannot really explain the observed results. Here are some comments to improve this manuscript:

1) No clear correlation could be determined between the viscosity, thermal stability and tensile properties of the various treatments used in this study. The results are all over the place, and even the concept of HBD and HBA cannot explain why certain trends are observed. Only the duration of dissolution seem to have any impact on the tensile properties; longer or incomplete dissolution (as seen in the case of GF, CCUF, and CCGF) seem to improve the tensile properties. It seems almost as if the authors have accidentally stumbled upon a method to improve the tensile strength and thermo-stability of the regenerated cellulose fibers in this study. It remains to be seen if the results can be replicated.

2) The introduction needs a caveat differentiating this study from the previous reports. Please cite or explain how the experimental set-up of this study is unique and has never been seen before.

3) Fiber production method: Although it is stated 3 times that a dry-jet wet-spinning method has been employed to produce the fibers, no drum rolls are depicted in Figure 1 that are necessary to draw and wind the extruded fibers. Without a proper spinning action, could it still be described as "wet-spinning" process? Seems more like anti-solvent coagulation or precipitation process.

As seen in Line 121, how were the fibers stretched? Was it done manually? Please be accurate in the method description. Also please depict the processes accurately in Figure 1.

4) In figure 1: Please use the term X to indicate magnification i.e., 100X for SEM images and 400X for optical microscope images. Also, the optical microscope image is totally dark, what were the authors thinking of depicting here?

5) In line 137: What kind of peak fitting function was used for the XRD peaks?

6) In Table 2: Did you mean Young's modulus instead of initial modulus?

7) The difference in % Crystallinity is clearly insignificant between all treatments, as per Table 2. Please stop using crystallinity as a point of reference to explain the various property changes, such as the tensile strength. Also, please stop pointing out that the crystallinity has "improved" by 2 or 3 percent, when it is clear that the so called "improvement" is statistically insignificant (both in the results and discussion and in the conclusion).

8) The pores cannot be seen clearly in the SEM images except for Figure 4C. Please use a better method to calculate porosity, such as BET analysis, if at all necessary. Or prepare better SEM images that show significant differences in surface characteristics. The current images CANNOT be used to discuss about porosity, unfortunately.

Author Response

Responses to Reviewer 2 Comments

Comment 1:

Although the authors have conducted a series of expected analytical studies to develop and justify a superior method for spinning fibers from pulp, there is still a lot left to be desired. Even the basic premise that whatever behavior is exhibited by the cellulose/[AMIM]Cl system could be explained away by using a linear HBD and HBA scale is faulty. It is generally known in the field those ionic liquids dissolve cellulose via the formation of electron acceptor AND electron donor complexes and not just the one thing or another (Cao et al., 2016, doi.org/10.1007/s00253-016-8057-8). The premise of this study simplifies the complex dissolution process, and therefore cannot really explain the observed results. Here are some comments to improve this manuscript:

Response 1:

Thank you for your suggestion. Indeed, we agree with you that there is still a lot to do for fully understand the process. The presented work is just using the AMIMCl system for example to illustrate the possibility to generated cellulose fiber with improved strength. Actually, in the present work we are not focus on the dissolution of the cellulose, but more focusing on the whole process. As we realized that the addition of HBA and/or HBD not only altered the dissolution of cellulose but also improved the physical strength of fiber with minor change in crystallinity.

The information you provided is very useful. Thereby, we have revised the introduction in the manuscript and added the mentioned literature (REF 9). Please find Line 44-48 and Line 50-52 in Page 1-2 for more details.

Comment 2:

1) No clear correlation could be determined between the viscosity, thermal stability and tensile properties of the various treatments used in this study. The results are all over the place, and even the concept of HBD and HBA cannot explain why certain trends are observed. Only the duration of dissolution seem to have any impact on the tensile properties; longer or incomplete dissolution (as seen in the case of GF, CCUF, and CCGF) seem to improve the tensile properties. It seems almost as if the authors have accidentally stumbled upon a method to improve the tensile strength and thermo-stability of the regenerated cellulose fibers in this study. It remains to be seen if the results can be replicated.

Response 2:

Thanks for your comments. Again, we agreed that there is still a lot to be done. We have amended the discussion to clarify the correlations. The key finding of the manuscript is that the addition of HBA/HBD in IL for cellulose dissolution caused a significate improvement in physical strength of regenerated fibers without major difference in crystallinity (Line 247-250 and Line 254-260). The crystallinity of all of the fibers was between 60% and 63%, and the tensile strength was improved by 136.62%. We think it is interesting to report this, as typically the physical strength is usually related to crystallinity.

A possible reason was the changes of hydrogen bonding, as the HBA/HBD in AMIMCl facilitated the crosslinking of the cellulose macromolecules in both the solution (viscosity) and regeneration (FTIR). We assumed that the HBD and HBA could adjustment of hydrogen bond distribution in ionic liquid during the cellulose dissolution and regeneration process.

In addition, we must announce that the experiment has been repeated several times in our lab to confirm the repeatability. But we are always happy to be connected if there is anyone are interested to repeat or carry on the study.

Comment 3:

2) The introduction needs a caveat differentiating this study from the previous reports. Please cite or explain how the experimental set-up of this study is unique and has never been seen before.

Response 3:

Thanks for your comments. The introduction was modified according to the suggestion, please check Line 68-81 in Page 2 for more details.

Comment 4:

3) Fiber production method: Although it is stated 3 times that a dry-jet wet-spinning method has been employed to produce the fibers, no drum rolls are depicted in Figure 1 that are necessary to draw and wind the extruded fibers. Without a proper spinning action, could it still be described as "wet-spinning" process? Seems more like anti-solvent coagulation or precipitation process.

As seen in Line 121, how were the fibers stretched? Was it done manually? Please be accurate in the method description. Also please depict the processes accurately in Figure 1.

Response 4:

Thanks for your comments. The cellulose solution is extruded through an extruder with a diameter of 0.8 mm, and then passed through 2 cm of air into a coagulation bath (deionized water). The wet cellulose fiber washed three times in the washing trough (length ×width× height is 50cm×30cm×30cm) at 1m/min and manually winded onto a 10 cm diameter straight glass tube, the stretch ratio is 1:1 during all the washing process. The regenerated cellulose fiber with the glass tube was soaked in deionized water for 24 h and then washed at 80°C and remove the solvents with room-temperature deionized water until the content of AMIMCl in the regenerated fiber was less than 0.3 wt%.

We have updated Figure 1 according to the suggestion, to give a better presentation. And the description of process (Line 131-135 in Page 4) was expanded to present more details of preparation of regenerated cellulose fibers. Please find the corresponding changes in Line 117-120 and 131-135 in Page 3-4 with yellow highlight.

Comment 5:

4) In figure 1: Please use the term X to indicate magnification i.e., 100X for SEM images and 400X for optical microscope images. Also, the optical microscope image is totally dark, what were the authors thinking of depicting here?

Response 5:

Thanks for your careful check, and sorry for the confusions that was made. The polarizing microscope image without bright spot in the vision field indicates the successful cellulose dissolution.

We have amended Fig 1 (Page 3) and also expanded the legend.  Please check Line 117-120 in Page 3 for more details.

  Comment 6:

5) In line 137: What kind of peak fitting function was used for the XRD peaks?

Response 6:

Thanks for your comments. The XRD data was treated according to the Segal methods with Gaussian fitting. The description was expanded and the related literature (REF 25,26) were added.

Comment 7:

6) In Table 2: Did you mean Young's modulus instead of initial modulus?

Response 7:

Thanks for your comments, but the results presented were initial modulus. The mothed and instrument we used in this manuscript gives initial modulus instead of Young's modulus. The use of initial modulus was referred to the literature method (REF 23,24).

Comment 8:

7) The difference in % Crystallinity is clearly insignificant between all treatments, as per Table 2. Please stop using crystallinity as a point of reference to explain the various property changes, such as the tensile strength. Also, please stop pointing out that the crystallinity has "improved" by 2 or 3 percent, when it is clear that the so called "improvement" is statistically insignificant (both in the results and discussion and in the conclusion).

Response 8:

Thanks for point this out. We agree with you that the key finding in this work should be the significate improvement in physical strength with only minor change in crystallinity. Therefore, the words indicating the changes of crystallinity were revised. The discussion relating to the XRD results in Page7-8 was modified accordingly. Please check Line 247-250, Line 254-260, Line 265-267 and Line 273-282 in Page 7-8 for more details.

Comment 9:

8) The pores cannot be seen clearly in the SEM images except for Figure 4C. Please use a better method to calculate porosity, such as BET analysis, if at all necessary. Or prepare better SEM images that show significant differences in surface characteristics. The current images CANNOT be used to discuss about porosity, unfortunately.

Response 9:

Thanks for your comments. We have replaced Fig.4 to made it more clearly and marked the irregular pores, protrusions and cracks with red highlight in the pictures. Instead of discuss of porosity, the morphology changes with additives should be smoother surface, and less protrusions and cracks. We have amended the corresponding paragraphs in results (Line 309-318 and Line 331-335, P11-12), please check the revised manuscript for more details.

In addition, we actually have done the BET test with analyzer (Quantachrome Instruments, Autosorb-iQ), using nitrogen gas adsorption at 77 K, the outgas tempture was 120°C. The total pore volume of BF, UF, GF, CCF, CCUF and CCGF was 0.02858, 0.02384, 0.01804, 0.02384, 0.02702 and 0.02542 cc/g respectively. The Barret-Joyner-Halenda theory desorption surface area was between 12-16 m2/g and   pore volume was all 0.02 cc/g. However, the BET result did not give much valuable information that closely related to the manuscript. Therefore, they are not included in the submitted manuscript.

Round 2

Reviewer 1 Report

I read carefully the improved version of the manuscript and I found that the authors managed to make better use of the results. However, a number of results that I previously reported remained unresolved / incorrect. I am referring to the part of FTIR where the authors failed to correct the errors. Crystalline indices must be calculated. The assigmenetns of the crystalline regions of cellulose are incorrect. The errors are still marked in Fig. 5.Please correct the errors. For the crystallinity indeces by IR spectroscopy I recommend the authors to consider the following paper :
https://doi.org/10.1016/j.carres.2005.08.007, where they can find the attribution of the crystalline in cellulose.
Based on these consideration, I recommend minor revision for the present form of the manuscript.

Author Response

Responses to Reviewer 1 Comments

Comment 1:

I read carefully the improved version of the manuscript and I found that the authors managed to make better use of the results. However, a number of results that I previously reported remained unresolved / incorrect. I am referring to the part of FTIR where the authors failed to correct the errors. Crystalline indices must be calculated. The assignments of the crystalline regions of cellulose are incorrect. The errors are still marked in Fig. 5. Please correct the errors. For the crystallinity indices by IR spectroscopy I recommend the authors to consider the following paper :

https://doi.org/10.1016/j.carres.2005.08.007, where they can find the attribution of the crystalline in cellulose.

Based on these consideration, I recommend minor revision for the present form of the manuscript.

Response 1:

Thank you for insisting the correction. We indeed learn a lot during the refining of FTIR result. The Figure 5 was revised and the discussion was almost rewritten. Please find the changes in Page 5 and Page 14-16.

Briefly, the crystallinity index obtained by FTIR according to Ref 28-29. It was evaluated from the ratios of the band at A(1427, 1425)/A(913, 897) and A(1368)/A(1262, 1260). The Calculation of the total crystallinity index (TCI) was performed from the ratio of the peaks A1372/ A2900 [Ref 29]. With these results, the FTIR discussion has been modified accordingly. Please check the revised manuscript with blue highlighted for more details.

Reviewer 2 Report

The authors have adequately addressed this reviewer's previous suggestions. However, there are new concerns as states below:

  • The FT-IR spectra looks unusual with peak maxima in both the positive and negative direction. If an ATR accessory was used as claimed, then proper pre-processing needs to be applied before the spectra could be interpreted.
    • Pre-processing steps namely baseline correction, mean normalization, and MSC scatter correction has to be applied.
    • Please edit the FT-IR discussion section after properly pre-processing the spectra.
  • Please check for grammatical errors. Wrong tenses are used throughout the manuscript.

Author Response

Responses to Reviewer 2 Comments

Comment 1:

The authors have adequately addressed this reviewer's previous suggestions. However, there are new concerns as states below:

The FT-IR spectra looks unusual with peak maxima in both the positive and negative direction. If an ATR accessory was used as claimed, then proper pre-processing needs to be applied before the spectra could be interpreted.

Pre-processing steps namely baseline correction, mean normalization, and MSC scatter correction has to be applied.

Please edit the FT-IR discussion section after properly pre-processing the spectra.

Please check for grammatical errors. Wrong tenses are used throughout the manuscript.

Response 1:

Thank you for point this out and also for the great suggestions. The FTIR results were re-processed according to Ref 27-29, and the discussion was corrected with the revised Figure 5. Please check the updated FTIR section in Page 5 and Page 14-16.

The FTIR method was corrected as follow: The Fourier-transform infrared (FT-IR) spectra of the regenerated cellulose fibers were recorded by an ALPHA FTIR spectrophotometer (Bruker Corporation, Billerica, MA, USA) with attenuated total reflectance (ATR) and diamond-ZnSe crystal. The wavelength ranged from 500 to 4000 cm−1 with a resolution of 4 cm−1. All of the spectra were corrected against an air background. The spectra of all samples were corrected by water vapor, automatic base-line correction, and smoothing by MONIC 9.0 software, and the spectra are presented without ATR correction [Ref 27]. The crystallinity index obtained by FTIR was calculated according to REF27-29.

And the results and discussion related to FTIR has been revised accordingly. Please check the revised manuscript with blue highlighted in Page 14-16 for more details.

Also, the manuscript has been proofread to correct the grammatical error, all the changes has been marked with red highlight. Please find the revised manuscript for more details.

Round 3

Reviewer 1 Report

The authors improved the paper according to the reviewer's suggestion and can be accepted in the present form.

Reviewer 2 Report

The authors have strived well to address this reviewer's concerns. The manuscript may be accepted after a quick spell-check.